# Ethyl Carbamate in Fermented Food Products: Sources of Appearance, Hazards and Methods for Reducing Its Content

**DOI:** 10.3390/foods12203816

**Published:** 2023-10-18

**Authors:** Maksim Yu. Shalamitskiy, Tatiana N. Tanashchuk, Sofia N. Cherviak, Egor A. Vasyagin, Nikolai V. Ravin, Andrey V. Mardanov

**Affiliations:** 1All-Russian National Research Institute of Viticulture and Winemaking “Magarach” of the Russian Academy of Sciences, 298600 Yalta, Russia; mshalamitskiy@yahoo.com (M.Y.S.); magarach_microbiol.lab@mail.ru (T.N.T.); sofi4@list.ru (S.N.C.); 2Institute of Bioengineering, Research Center of Biotechnology of the Russian Academy of Sciences, 119071 Moscow, Russia; egor.vasyagin@gmail.com (E.A.V.); nravin@biengi.ac.ru (N.V.R.)

**Keywords:** ethyl carbamate, wine, yeast, lactic acid bacteria, fermented foods

## Abstract

Ethyl carbamate, the ethyl ester of carbamic acid, has been identified in fermented foods and alcoholic beverages. Since ethyl carbamate is a probable human carcinogen, reduction of its content is important for food safety and human health. In alcoholic beverages, ethyl carbamate is mostly formed from the reaction of ethanol with urea, citrulline and carbamyl phosphate during fermentation and storage. These precursors are generated from arginine metabolism by wine yeasts and lactic acid bacteria. This review summarizes the mechanisms of ethyl carbamate formation, its impact on human health and methods used in winemaking to minimize its content. These approaches include genetic modification of *Saccharomyces cerevisiae* wine strains targeting pathways of arginine transport and metabolism, the use of lactic acid bacteria to consume arginine, direct degradation of ethyl carbamate by enzymes and microorganisms, and different technological methods of grape cultivation, alcoholic fermentation, wine aging, temperature and duration of storage and transportation.

## 1. Introduction

Wine is a traditional fermented alcoholic drink that has been known for thousands of years. Wine is rich in valuable compounds, such as polyphenols and vitamins, that can benefit human health [1,2,3]. Red wine polyphenols in particular exhibit anti-inflammatory, cardioprotective, antiallergic and antitumor effects [1,4]. Drinking wine as an aperitif stimulates gastric secretion and improves digestion [5]. However, the composition of wine is also represented by carbonyl compounds, including acetaldehyde, acrolein, formaldehyde, ethyl carbamate and furfural [6], high concentrations of which may be the key cause of carbonyl stress, which contributes to aging, liver damage, and the development of neurodegenerative diseases and other disorders such as diabetes and kidney failure [7,8].

One of the main indicators of the safety of wine for human health is the content of ethyl carbamate. There is evidence that the average amount of ethyl carbamate ingested from food is about 15 ng/kg of body weight per day. These values were calculated for food items including bread, dairy products and soy sauce, while alcoholic drinks were not taken into account [9,10]. Concerns about ethyl carbamate content and toxicity in regularly consumed foods and alcoholic beverages have sparked a global interest in assessing possible risks to human health.

The Joint FAO/WHO Expert Committee on Food Additives (JECFA) assessed exposure to ethyl carbamate in 2005 and concluded that it is not of particular concern in food intake [11], with the exception of alcoholic beverages, in which ethanol contributes to the carcinogenicity of ethyl carbamate [12,13,14]. It is also known that the combination of alcoholic beverages and food can increase the potential carcinogenic risk of ethyl carbamate by about four times [11,15].

The content of ethyl carbamate in food products varies over a fairly wide range: the highest level, up to 344 μg/kg, was found in Chinese red sufu, fermented soy curd [15]. Baking bread resulted in an increase in ethyl carbamate by 3–8 times, which in quantitative terms ranged from 3.5 to 33.8 µg/kg of fresh weight. It is caused by the breakdown of azodicarbonamide, which is used as a dough improver [15,16,17]. The mass concentration of ethyl carbamate in soy sauce can reach 108 µg/kg.

Ethyl carbamate has been identified in alcoholic beverages such as wine, beer, and yellow rice wine, as well as fermented foods such as bread, yogurt, soy sauce, vinegar, and soy paste [9,10,15,18,19,20,21,22,23,24,25,26]. The content of this component in alcoholic beverages also varies widely: from 8 µg/L in dry white wine to 111 µg/L in sake [27], and from 100 to 22,000 µg/kg in stone-fruit brandy [28]. The minimum content of ethyl carbamate among alcoholic beverages was found in beer, from 0 to 5.8 µg/kg [15]. Chinese rice wine contains almost twice as much ethyl carbamate as other Chinese alcoholic beverages [29]. An analysis of the data from the literature indicates that, in general, the content of ethyl carbamate in fermented food products is significantly lower than in alcoholic beverages, which may be due to the different ethanol content [15].

The presence of ethyl carbamate has been established in grape wines from Brazil, Germany, Hungary, Portugal, Australia and other countries. Despite this, there is currently no international standard that regulates the maximum allowable level of ethyl carbamate in foods. However, some countries, such as Canada, South Korea, and certain European Union member states (e.g., France, Germany, and the Czech Republic), have set limits on ethyl carbamate content in alcoholic beverages [30]. Thus, the maximum allowable concentration of ethyl carbamate in wine in Canada and the Czech Republic is 30 µg/L, while in the USA it is legally regulated at 15 µg/L [8]. Based on studies on the toxicity of ethyl carbamate, for the first time in 1985, Canadian authorities introduced a law restricting this substance in alcoholic beverages: 30 µg/L for table wines, 100 µg/L for fortified wines and 400 µg/L for fruit brandies and liqueurs [15,26]. Health Canada has also set maximum ethyl carbamate levels for other alcoholic beverages such as sake and liquor [31]. Canadian regulations have been adopted by other countries such as the Czech Republic, Brazil, France, Germany and Switzerland. South Korea has set a maximum limit (30 µg/L) for ethyl carbamate in table wines only. The Food and Drug Administration (FDA) has notified all countries that export wines to the USA that the ethyl carbamate content in fortified wines should not exceed 60 µg/L [15,26,27]. The maximum allowable content of ethyl carbamate in distilled spirits in Canada, the Czech Republic and France is 150 µg/kg [15].

## 2. Characteristics of Ethyl Carbamate and Its Impact on Human Health

Ethyl carbamate (C_3_H_7_NO_2_, CAS No. 51-79-6), also known as urethane, is the ethyl ester of carbamic acid. In the 1940s, it was widely used in medicine as a sleeping pill for people and as an anesthetic for animals. However, in 1943 the drug was found to be carcinogenic in a number of animals such as mice, rats, hamsters and monkeys [13,32]. Based on clinical trials, the International Agency for Research on Cancer (IARC) classified ethyl carbamate as a possible human carcinogen (Group 2B) in 1974 [33]. Later in 2007, IARC upgraded it to a probable human carcinogen (Group 2A) [15,34].

Ethyl carbamate is rapidly absorbed from the gastrointestinal tract and skin, and then distributed throughout the body [35,36,37]. Up to 90% of the absorbed substance is hydrolyzed by microsomal esterase in the liver and excreted as ethanol, carbon dioxide and ammonia. About 5% of ethyl carbamate is excreted in the urine after hydroxylation and conjugation (Figure 1).

The presence of a carbonyl group in the structure of ethyl carbamate and its derivatives makes these compounds electrophilic and, therefore, highly reactive towards cell nucleophiles, which is the main cause of various diseases and disorders [7,8,38]. In addition, carbonyl compounds cause an increase in reactive oxygen species and can trigger apoptosis in addition to hepatocellular adenoma and carcinoma due to chronic hepatotoxicity.

Ethyl carbamate is also oxidized to vinyl carbamate by cytochrome P450 2E1 and then converted to vinyl carbamate epoxide, which can covalently bind to DNA, RNA, and proteins leading to the formation of adducts able to cause a carcinogenic effect (Figure 1) [37,39]. The carcinogenic potential of ethyl carbamate lies in the induction of mutations, which contributes to the progression of precancerous cells into a malignant form, increasing the risk of developing cancer of the lungs, lymph, liver and skin [8,12,13,14,39,40,41,42,43].

## 3. The Role of Wine Microorganisms in the Formation of Ethyl Carbamate

Ethyl carbamate is formed as a result of the reaction of ethanol and urea, cyanate, citrulline (Cit), carbamyl phosphate or other N-carbamyl compounds (Figure 2) [10,15,20,26,44]. The main source of ethyl carbonate in fermented foods is the reaction between urea and ethanol, and the high ethyl carbonate content in stone fruit spirits is mainly due to the cyanogenic glycosides present in their pits [30]. The most important ethyl carbamate precursors in wine are urea, citrulline and carbamyl phosphate [20,24,44,45].

In wine, ethyl carbamate is formed as follows: arginine, usually one of the most abundant amino acids available to yeast in grape juice, undergoes enzymatic hydrolysis in the early and middle stages of alcoholic fermentation to form urea [44,46]. Specifically, arginase (EC 3.5.3.1) converts L-arginine into L-ornithine and urea. When urea cannot be further metabolized in the cell and accumulates above a critical concentration, the yeast secretes it into the wine during or at the end of fermentation. Subsequently, urea reacts with ethanol to form ethyl carbamate (Figure 2).

Citrulline is already present in grape must, but it can also be formed as a result of arginine anabolism by the yeast *Saccharomyces cerevisiae*, when ornithine and carbamoyl phosphate react to form arginine, with citrulline acting as an intermediate [44]. Ethyl carbamate can also appear in wine as a result of the ethanolysis with citrulline and carbamyl phosphate, formed as a result of the degradation of arginine with the participation of certain strains of lactic acid bacteria (LAB) and yeast (Figure 2).

Ethyl carbamate is also formed in distilled ethyl alcohol, especially obtained from stone fruits (cherries, apricots or plums), as a result of the reaction of ethanol and isocyanate, a by-product of the enzymatic hydrolysis of cyanoglycosides present in these fruits [9,10,15,24,30,47,48]. The amount of ethyl carbamate in distilled spirits is influenced by the distillation temperature, ethanol content and the design of the distillation apparatus [30]. The content of this compound during exposure and storage can increase significantly. About 80% of the ethyl carbamate present in alcohols is formed during the distillation step and/or during the first 48 h after distillation [15].

Yeast strains can differ in their ability to catabolize arginine and urea during fermentation. Yeasts that secrete a large amount of urea into the medium usually have a high ability to break down arginine to urea, but a weak ability to hydrolyse it, which may be the result of low urea amidolyase activity, its inhibition by high ammonia content, a deficiency of cofactors necessary for enzyme action or hyperactivity of arginase. Genetic factors, as well as environmental factors such as pH and temperature, also affect the amount of urea released by cells [24,46,49]. An increase in the content of arginine in wine may be due to the effect of ethanol on the porosity of yeast membranes [49] or yeast autolysis [50].

When studying the effect of yeast strains on the formation of ethyl carbamate in alcoholic products, it was found that the *S. bayanus* yeast produces significantly fewer ethyl carbamate precursors than *S. cerevisiae* [49]. In wines obtained with the use of the yeast *Wickerhamomyces anomalus*, higher urea content (910 μg/L) was noted than with *S. cerevisiae* (300 μg/L). Moreover, the concentration of urea increased to 1261 μg/L with the joint introduction of the yeast of these two genera. At the same time, the transcriptional activity of the arginase gene (*CAR1*) in *W. anomalus* increased by 140%, while in *S. cerevisiae* it decreased by 40%.

There is evidence that representatives of the non-*Saccharomyces* yeast of the genera *Pichia*, *Schizosaccharomyces* and *Zygosaccharomyces* can be active regulators of the formation of urea from arginine. Some members of the genera *Schizosaccharomyces* and *Zygosaccharomyces* can produce less urea and decompose it more efficiently than *S. cerevisiae* [51]. The gene coding for ATP-independent urease (*URE*) was mainly found in the yeast of the genus *Schizosaccharomyces*. Another urea degradation pathway (involving urea carboxylase and allophanate hydrolase) has been identified predominantly in yeasts of the genus *Pichia* and *Zygosaccharomyces*. The authors point out that the use of such yeast allowed for the reduction of the urea content during fermentation [51].

### 3.1. Ethyl Carbamate in Wine: The Role of Lactic Acid Bacteria

Despite the fact that urea produced by yeast is the main precursor of ethyl carbamate in wine, LAB can also contribute to its formation [52]. Arginine is available for both wine yeast and LAB metabolism during malolactic fermentation (MLF), which usually proceeds after alcoholic fermentation. Depending on the LAB strain used, the amount of ethyl carbamate in wine may slightly increase [53].

The catabolism of arginine by LAB via the arginine deiminase (ADI) pathway leads to the production of ammonia, ornithine and CO_2_ (Figure 3). This pathway is the most common for arginine degradation under anaerobic conditions [54]. The first enzyme of this pathway, arginine deiminase (ArcA), is responsible for the degradation of arginine to form citrulline, which can react with ethanol to form ethyl carbamate (Figure 3) [55,56]. The formation of ammonia from arginine contributes to an increase in the pH of the wine, which may be physiologically significant for the adaptation of LAB to a low pH value. It has been shown that the ADI pathway is an important source of energy for bacterial growth, and LAB strains capable of obtaining energy from arginine catabolism may be more competitive in wine than strains unable to degrade arginine [57,58]. There are reports that the ADI pathway is sometimes indirectly associated with the production of biogenic amines, especially putrescine, because this amine can be formed from ornithine with the participation of LAB [59].

In one of the studies, the presented assessment of the ability of industrial acid-reducing strains to secrete and use citrulline confirmed that LAB secretes citrulline as a result of arginine degradation. At the same time, in an arginine-rich medium, citrulline was retained in the cell during cell growth and released during cell lysis. All strains have been found to utilize citrulline, and some of them are able to recycle previously secreted citrulline [60].

LAB strains are distinguished by their ability to degrade arginine, but there is little information on the potential of LAB species to degrade arginine under winemaking conditions. Some strains are able to form small amounts of citrulline from arginine. Uncontrolled enrichment of the must with nitrogenous substances can increase the likelihood of developing LAB at the end of fermentation, which can lead to an increase in the level of ethyl carbamate even at optimal temperatures during long-term storage.

An alternative way to prevent the formation of ethyl carbamate in wine is the use of *Lactobaccilus hilgardii* and *Oenococcus oeni* to utilize arginine and citrulline [15,61]. The presence of these bacteria in the starter cultures reduced the ethyl carbamate content in the fermented food products [61]. In the second work, LAB was used as starter cultures for malolactic fermentation after the end of alcoholic fermentation [15]. Generally, all heterofermentative wine LABs, including strains of *L. buchneri* and *L. brevis* [56,62,63] and *O. oeni* [54] hydrolyze arginine. There are reports showing that some representatives of *L. plantarum* consume arginine through the ADI pathway [64]. Araque et al. [65] found a correlation between the presence of genes of the ADI pathway and the ability to degrade arginine by LAB isolated from wine. This feature was also noted for all strains of heterofermentative lactobacilli, *O. oeni*, *Pediococcus pentosaceus* and some strains of *Leuconostoc mesenteroides* and *L. plantarum* [54,65]. There is evidence that during alcoholic fermentation, some strains of *Lactobacillus* decomposed arginine simultaneously with yeast, which led to a decrease in the production of urea by yeast [51].

### 3.2. Genetic Modification of Wine Yeast to Reduce the Formation of Ethyl Carbamate

Selection of industrial yeast strains based on the application of a complex of modern techniques and approaches of postgenomic biology, metabolic engineering and genome editing are relevant for winemaking. Modification of yeast wine strains by genetic techniques provides significant progress in the inhibition of urea metabolism.

Import of arginine from the culture medium into the yeast cell is primarily enabled by the Can1 arginine amino acid transporter localized to the plasma membrane of yeast (Figure 4) [66]. In addition, general amino-acid permease Gap1 is responsible for the transport of arginine and related compounds such as ornithine and citrulline into the cell. Arginase, encoded by the *CAR1* gene, cleaves arginine to urea, which is either excreted by *DUR4* permease or metabolized by urea amidolyase (*DUR1,2*), responsible for the conversion of urea to ammonia and carbon dioxide. Active transport of urea from the medium into the yeast cells is performed by the urea-proton symporter (*DUR3*) (Figure 4). Thus, inhibition of *CAR1* expression and increased expression of *DUR1,2* and *DUR3* leads to a decrease in the concentration of urea in the medium and, consequently, to a decrease in the formation of ethyl carbamate [20,67,68,69].

Despite numerous and successful attempts to improve the characteristics of wine yeast strains by genetic engineering, only two of them, strain ML01 carrying genes of the MLF pathway [70] and strain ECMo01 [61], are officially registered for use in the USA, Canada, and Moldova. Wider use of GM strains is constrained by both the conservatism of winemakers and well-known public prejudice and legislative restrictions in the field of genetic engineering in the food industry [71]. Many studies were aimed at obtaining yeast strains with low ethyl carbamate production in wine [61]. The ECMo01 strain contains an additional copy of the *DUR1,2* amidolyase gene under the control of regulatory sequences of the phosphoglycerate kinase *PGK1*. In this strain, the expression of the *DUR1,2* gene was increased by 17 times, which led to a decrease in the concentration of urea, and the produced ammonia was used as a source of nitrogen. The concentration of ethyl carbamate in wine obtained using the ECMo01 strain was reduced by 90%, while the strain was equivalent in its phenotypic characteristics to the parental strain 522. Subsequently, the obtained wine yeast strain 522DUR3 with elevated expression of the *DUR3* transporter gene enabled an 81% reduction in ethyl carbamate content in wine [25].

Genetically engineered yeast strains with a null mutation in *CAR1* were specially adapted to reduce the concentration of ethyl carbamate in sake, cherry spirits and Chinese rice wine. There were no significant differences in yeast fermentation activity compared to the corresponding parental strains [72].

To reduce the production of urea, a recombinant strain YZ22 (*Δcarl*/*Δcarl*) with arginase deficiency was obtained from diploid wine yeast strain WY1 by successive deletion of two *CAR1* alleles. The results of RT-qPCR showed that strain YZ22 almost did not express the *CAR1* gene, and the arginase activity was 12.6 times lower than that of the parental strain WY1. According to the results of alcoholic fermentation, it was found that the content of urea and ethyl carbamate in wine decreased by 77.9 and 73.8%, respectively. In addition, the lower urea content of the wine resulted in slower and quantitatively less formation of ethyl carbamate during storage. The resulting strain in fermentation activity was equivalent to the parental strain [72].

The use of CRISPR/Cas-mediated genome editing is a promising technique for creating improved yeast strains since it does not bear the risks associated with the introduction of foreign genes and genetic elements, markers of antibiotic resistance, into the genome of yeast food strains, i.e., the obtained strains are safe, according to the regulatory restrictions adopted in some countries. The CRISPR/Cas9 system was used to make wine strains with reduced urea production. Derivatives of wine strains EC1118 and AWRI1796, defective in both alleles of the *CAN1* gene, were obtained [73].

Another way to reduce urea production is to inactivate transporters responsible for the uptake of arginine into yeast cells from the culture medium. Modified wine yeast strains with alterations in arginine catabolism pathways have already been developed for the production of sake and sherry, but there is no information on the role of *CAN1* in urea production during yeast fermentation. The resulting recombinant strains ScEC1118can1 and ScAWRI796can1 with eliminating the *CAN1* arginine permease pathway were characterized by reduced urea production (by 18–36% compared to the original ones) under the conditions of experimental winemaking with the preserved ability to ferment the synthetic substrate, although with a slightly reduced growth rate [73]. The advantage of introducing a mutation in the *CAN1* gene compared to other methods of modifying arginine utilization pathways is that this technique is less sensitive to fluctuations in the content of nitrogen sources in the composition of the wort and has less effect on the growth parameters of yeast strains.

One more way to minimize ethyl carbamate formation is to reduce the concentration of urea by converting it into ammonia and CO_2_ by the bifunctional enzyme urea amidolyase encoded by *DUR1,2* [61,74,75]. Gene *DUR1,2* is repressed when the preferred nitrogen sources of yeast are available. Improving urea degradation through the constitutive expression of *DUR1,2* in yeast has been achieved and the reduction of ethyl carbamate in wine and sake was 89.1% and 68.0%, respectively [25,61]. Wu D. et al. developed a *DUR1,2* expression cassette to insert the strong *PGK1* promoter upstream of the *DUR1,2* gene loci in the wild-type industrial yeast *N85DUR1,2* and *N85DUR1,2-c* with deleted *CAR1* genes [76]. During small-scale fermentation, the concentrations of urea and ethyl carbamate were reduced by engineered strain *N85DUR1,2* by 75.6% and 40.0% compared to the parental yeast strain, respectively. The *N85DUR1,2-c* strain reduced urea and ethyl carbamate by 89.1% and 55.3%, respectively [76].

Recently, a CRISPR/Cas9-based genome editing was used to modify a polyploid wild yeast *S. cerevisiae* strain used for the production of Korean traditional rice wine [77]. In addition to the *CAR1* gene, the *GZF3* gene encoding the transcription factor controlling expression levels of genes *DUR1,2* and *DUR3* was deleted to further reduce the formation of ethyl carbamate. The use of a double deletion strain for wine brewing resulted in a significant reduction of the ethyl carbamate content in this wine when compared to the wild-type strain without affecting fermentation capability.

## 4. Technological Factors Affecting the Formation of Ethyl Carbamate in Wine

The level of ethyl carbamate in alcoholic beverages is significantly affected by a number of factors, including technological methods of grape cultivation, variety characteristics, genetic traits of yeast, temperature and duration of storage, and exposure to sunlight [10,19,28,46,78,79].

The rate of formation and quantity of ethyl carbamate depend on the ease of transfer of the carbamyl moiety and, in complex solutions, on the presence or absence of competing acceptor molecules [20]. It was shown that the formation of ethyl carbamate is proportional to the initial concentrations of urea, citrulline and ethanol [14,20,23,24,47].

The concentration of nitrogen components, such as arginine in juice and urea in wine, increases according to the amount of nitrogen fertilizer in the vineyard [10,26,46]. If the concentration of arginine in the juice exceeds 1000 mg/L, the vineyard should be considered over-fertilized [46].

Grapes varieties differ significantly in their ability to assimilate nitrogenous and mineral components, and therefore the amino acid profile of the must can vary significantly. The nitrogen status of a grape plant is determined by the root system and varies depending on the rootstocks used. The difference in the content of total nitrogen in grape inflorescences reaches 40%, while the level of nitrate nitrogen can vary by 10–12 times, depending on the combination of rootstock and scion used.

Uthurry et al. (2004) found ethyl carbamate concentrations in young wines to be between 1 and 10 µg/L [80]. Old, aged wines tend to show higher levels of ethyl carbamate, as ethyl carbamate forms slowly during aging [81].

A common technique in winemaking is the aging of wine on yeast precipitate after primary fermentation in order to form the organoleptic properties of wine. It has been found that in wines obtained from grapes with a low concentration of amino acids, after prolonged contact with the yeast precipitate, the concentration of ethyl carbamate was not increased. At the same time, there are no data to assess the effect of aging wine made from grapes with an excess content of assimilable nitrogen on the concentration of urea. Similarly, there is no information in the literature on the changes in the concentration of urea and other precursors of ethyl carbamate during the long-term aging of sparkling wines [46].

It was shown that when aging yellow rice wine with an alcohol content of 40% in bottles for 82 days, the amount of formed ethyl carbamate exceeded by 3.1 times the content of this component in wine with 20% alcohol (591 µg/kg and 189 µg/kg, respectively) [21]. On the contrary, no significant correlations were found between the formed ethyl carbamate (both in absolute content and per unit of urea) and the alcohol content in grape wines [10].

The type of aging also plays an important role in the formation of ethyl carbamate: when the process was carried out in oak barrels, the content of ethyl carbamate in wine was higher (57.4 µg/L) than when using stainless steel tanks (47.3 µg/L). This difference may be due to favorable conditions for the development of microorganisms during the aging of wine in oak barrels due to limited oxygen access and, consequently, an increase in the concentration of ethyl carbamate precursors [8,82].

Temperature and storage time are among the main factors that have a significant impact on the final concentration of ethyl carbamate and its precursors in wine [24,30,45,47]. With an increase in temperature by 10 °C, the formation of ethyl carbamate from urea and citrulline increases by 1.5–2 and 1.5–2.5 times, respectively [14].

Regarding the dynamics of the level of ethyl carbamate during storage, there is no consensus view in the literature. For example, the change in the content of ethyl carbamate in red wine during storage from 1997 to 2001 did not reveal a linear relationship: the ethyl carbamate concentration first decreased and then increased over the next two years [22]. At the same time, in wines from yellow rice, the ethyl carbamate content increased during storage for the first 200 days and reached 421 and 378 μg/kg by the end of the experiment (600 days) in bottled and packaged wines, respectively.

At the same time, during the storage of wine, the urea content decreases logarithmically. The rate of decrease in the concentration of urea is very high at the initial stage of storage, but slows down over time [14]. Storage temperature also plays an important role in the formation of ethyl carbamate. The chemical reaction between urea and ethanol increases exponentially with temperature [46]. Thus, in rice wine stored for 400 days at 37 °C, the mass concentration of ethyl carbamate was 7 times higher than in wine stored at 4 °C (509 and 84 μg/L, respectively) [21].

Storage of grape wines at temperatures below 12 °C did not significantly affect the formation of ethyl carbamate, regardless of the initial urea content [10]. At the same time, the content of ethyl carbamate increased from 15 to 30 μg/L in wine with a mass concentration of urea of more than 20 mg/L after less than 5 days of storage at 40 °C [10]. With an increase in the storage time to 152 days, the mass concentration of ethyl carbamate increased to 1136 µg/L. It was found that during storage, the total amount of ethyl carbamate formed was approximately 3.4% of the initial amount of urea.

Thus, special attention must be paid to the temperature regime during the storage and transportation of wine, since with a significant urea content, the ethyl carbamate level may exceed the recommended thresholds established in some countries within only a few days [10,30]. Preventive measures in this direction are minimizing the level of urea in wine and controlling the temperature during storage and transportation.

Among other factors, the formation of ethyl carbamate in wine is significantly influenced by the cooling time in the fermentation process, exposure to light, and storage time [14,28,30,78]. At the same time, no significant effect of wine pH on the rate of ethyl carbamate formation was found [14,24].

The inhibitory effect of glucose and fructose in model systems with urea and citrulline on the formation of ethyl carbamate was established. The presence of sugars in the solution with urea provided a decrease in the ethyl carbamate content by 11–26%, and in systems with citrulline by 20–26% compared with the control variant without the addition of sugars. No significant differences were found in terms of the effect between glucose and fructose, although lower ethyl carbamate values were noted in the variants of the experiment with fructose [26].

The important role of amino acids in the ethyl carbamate content was shown. After keeping the model systems at temperatures of 70 °C and 45 °C, the concentration of ethyl carbamate in the first variant of the experiment increased in the presence of citrulline by 7 times, in the presence of urea by 4 times, and in the presence of arginine by 2 times [26]. Under the same conditions, urea turned out to be a precursor with a higher rate of ethyl carbamate formation, on average 5 times higher than citrulline and 201 times higher than arginine. The higher reactivity of urea compared to citrulline has also been demonstrated by Stevens et al. and Hasnip et al. [23,24].

The formation of ethyl carbamate in fortified wines was studied in model systems with accelerated aging at a temperature of 45 °C for 4 months and at 70 °C for 1 month [26]. It has been shown that arginine can induce ethyl carbamate formation upon accelerated aging, without prior enzymatic metabolism by microorganisms, as previously thought [9,44,48]. Although much less reactive, arginine plays a significant role in fortified wines, as young wines can retain a high concentration of arginine [26]. Thus, the need for industrial control of nitrogen sources during fermentation is increasing in order to prevent excessive arginine content.

It has been found that the technological method of fortification provides a decrease in the mass concentration of urea in wine by 5–33%. The established trend was noted in 8 of the 12 fortified wines studied [26]. A decrease in the concentration of citrulline by 11–46% was noted in 7 out of 12 samples. This fact may be due to the dilution caused by the introduction of ethanol, or a decrease in the solubility of urea and citrulline with an increase in the concentration of ethanol, as was previously demonstrated [83,84]. At the same time, in relation to the quantitative content of arginine, an opposite trend was noted: its content increased by 2.9 times. An increase in the content of arginine in wine may be due to the effect of ethanol on the porosity of yeast membranes [49] or yeast autolysis [50].

## 5. Preventive Measures to Reduce the Content of Ethyl Carbamate in Wine Products and Direct Degradation of Ethyl Carbamate

In recent years, the consumption of alcoholic beverages, including wine, has grown significantly in the world, which has led to increased attention to the quality and safety of products. At the same time, special attention is paid to the content of ethyl carbamate in these products [14,80,85]. To date, many methods have been proposed to reduce the level of ethyl carbamate in alcoholic beverages, based mainly on either inhibition of the production of its precursors or their degradation, as well as the use of antioxidants (potassium metabisulfite), diammonium phosphate, and genetically modified yeast strains [15,70,86]. As a preventive measure, it was recommended to remove seeds from stone fruit, as well as strictly control the temperature regimes during storage and transportation of wine products [23]. As additional methods, it is proposed to maintain a low pH of the medium [61], filter wine after alcoholic fermentation, minimize exposure to light and/or reduce exposure or storage time [15,18,23,87].

Since urea is the main precursor of ethyl carbamate in wine, its hydrolysis into carbon dioxide and ammonia is a promising way to reduce the formation of ethyl carbamate [87,88]. Urease enzymes are marketed and approved by BATF for wine processing. However, urease activity is severely limited under wine conditions, especially at low pH and high ethanol content. Urease is inhibited by high concentrations of malic acid and fluorine residues (appearing from the use of cryolite in the vineyard) in excess of 1 mg/L. Any combination of these factors makes it nearly impossible to achieve the desired low urea levels even at very high enzyme dosages. Reduction of ethyl carbamate to trace amounts by this method is impossible [46,87,88].

Methods of direct degradation of ethyl carbamate in wine products based on the use of urethanase and microorganisms possessing this activity have also been proposed [89,90,91]. For example, *Lysinibacillus sphaericus* MT33 and *Candida ethanolica* strains J1 and J116 can reduce ethyl carbamate concentrations in Chinese liquors [92,93].

## 6. Determination of Ethyl Carbamate Content in Fermented Foods

Over the past decades, a number of analytical methods for the determination of ethyl carbamate in various fermented foods have been developed. The most common methods are gas chromatography–mass spectrometry (GC-MS) and high-performance liquid chromatography (HPLC) coupled with a fluorescence detector (FLD) [94,95]. The GC-MS is the most robust method which is adopted by the Association of Official Analytical Chemists International (AOAC International) as an official method for the detection of ethyl carbamate in fermented foods with limits of quantitation for table wine from 10 μg/L to 50 μg/L for distilled spirits [30,96]. However, this method requires preliminary preparation of samples by solid-phase extraction.

HPLC-FLD methods are generally less time-consuming and less expensive than GC-MS, but often require derivatization to enhance the analyte signal, and the presence of interfering compounds can lead to overstated results [95,97,98]. Several authors have reported that the sensitivity of LC-FLD was comparable to that of GC-MS [97,99].

Alternative methods for the detection of ethyl carbamate are flow-injection mass spectrometry, enzyme-linked immunosorbent assay, infrared spectroscopy, surface-enhanced Raman spectroscopy, and nanosensors. These methods could be used for a high-throughput screening without expensive analytical equipment, but they need to be optimized and require extensive validation to ensure that analytical results are reproducible [100,101,102,103,104].

## 7. Conclusions

Control and regulation of ethyl carbamate content is an urgent task to improve the safety of fermented foods, in particular wines. Since the main precursors for the formation of ethyl carbamate are urea and citrulline formed during the metabolism of arginine, reducing the concentration of these substances in the initial product makes it possible to regulate the formation of ethyl carbamate. At the stage of grape cultivation, controlled application of nitrogen fertilizers in vineyards is advisable. At the grape processing stage, the addition of arginine and urea as nitrogen sources should be avoided. To obtain fermentation products with a low content of ethyl carbamate, an effective approach is the use of genetically modified yeast strains producing a low level of urea. An alternative way to prevent the formation of ethyl carbamate in wine is the use of LAB, either simultaneously with yeast or after alcoholic fermentation, for the utilization of arginine and citrulline. Reducing the concentration of urea and ethyl carbamate in fermentation products can be achieved by enzymatic digestion.

Today, the official international method for the analysis of ethyl carbamate content in alcoholic beverages is gas chromatography–mass spectrometry. Other methods include high-performance liquid chromatography, enzyme-linked immunosorbent assay, infrared spectroscopy, surface-enhanced Raman spectroscopy and nanosensors.

## Figures and Tables

**Figure 1 foods-12-03816-f001:**
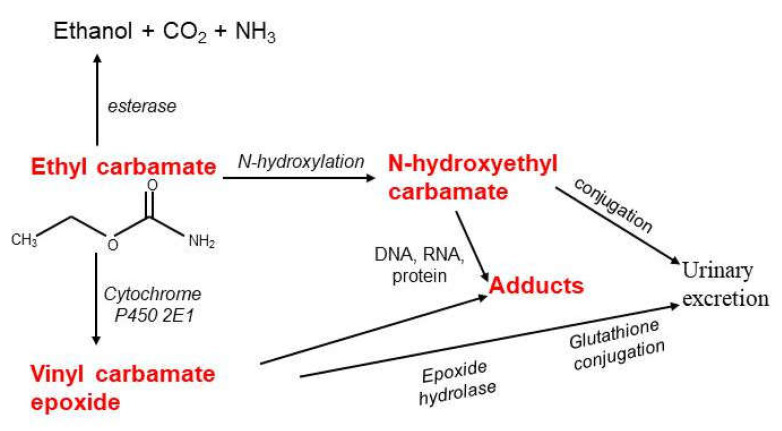
Probable pathways of ethyl carbamate metabolism. Toxic metabolites are marked in red.

**Figure 2 foods-12-03816-f002:**
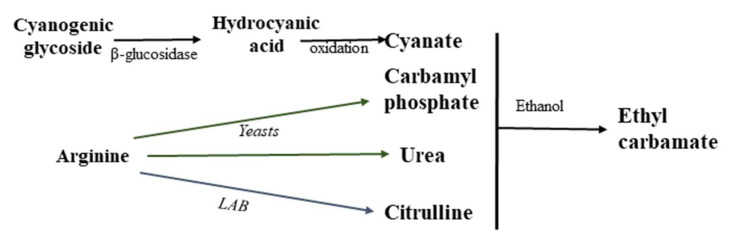
Mechanism of ethyl carbamate formation in alcoholic beverages.

**Figure 3 foods-12-03816-f003:**
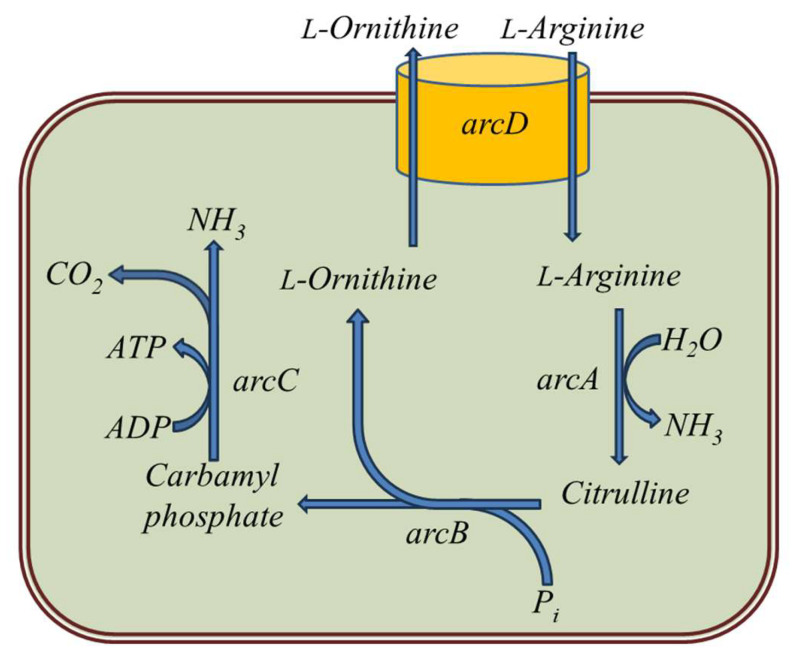
Arginine deiminase pathway in LAB. The enzymes of this pathway are (1) *arcD*—arginine: ornithine antiporter; (2) *arcA*—arginine deiminase (ADI); (3) *arcB*—ornithine transcarbamylase (OTC); and (4) *arcC*—carbamate kinase (CK).

**Figure 4 foods-12-03816-f004:**
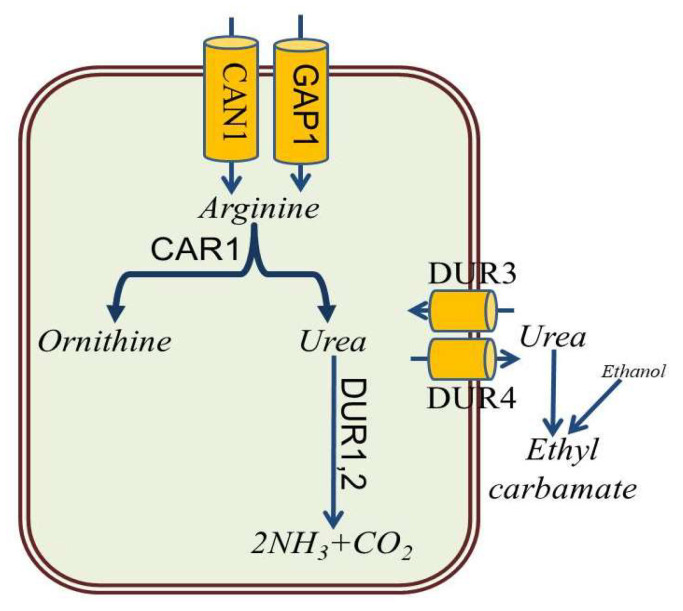
Main metabolic pathways of arginine degradation in *S. cerevisiae*.

## Data Availability

The data presented in this study are contained within the article.

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
