# Peer review of "Ethyl Carbamate in Fermented Food Products: Sources of Appearance, Hazards and Methods for Reducing Its Content"

_foods, 2023, doi:10.3390/foods12203816_

Round 1
Reviewer 1 Report
This was an interesting read and I think with a bit of tweaking it could be accepted.
Major point: the first paragraph, where a generic introduction is given on the 'benefits' of wine is extremely tedious and sould be rephrased. Phrases like improve the metabolism or biological value does not belong in scientific literature.
Some of thie figures lack proper citations especially fig 3 and 4
Please write the conclusions in full sentences
-please remove all 'etc.' in the text. In scientific writing one should not give readers the suspense of guessing with it meant with etc
-there are some cases where the font type seems different than the rest like in line 55 (it looks compressed)
Fig 3 and 126, please make the L- two fontsize smaller than the rest of the text
line 159, i think it is arginase
line 163 In some cases like here the whole name was given as in text citation (Hai Du). Another example is line 360
Line 174. I think this is the first mention of lactic acid bacteria but further on on LAB is used. PLease abbreviate it here so readers will know what is meant.
take care that all the gene names are in cursive like in line 260
Some references are not according to the guidelines like 9, 22, 77,93
Author Response
Comments and Suggestions for Authors
This was an interesting read and I think with a bit of tweaking it could be accepted.
Major point: the first paragraph, where a generic introduction is given on the 'benefits' of wine is extremely tedious and sould be rephrased. Phrases like improve the metabolism or biological value does not belong in scientific literature.
RE: We shortened and modified this section.
Some of thie figures lack proper citations especially fig 3 and 4
RE: Corrected
Please write the conclusions in full sentences
RE: The conclusion section has been rewritten
Comments on the Quality of English Language
-please remove all 'etc.' in the text. In scientific writing one should not give readers the suspense of guessing with it meant with etc
RE: Corrected
-there are some cases where the font type seems different than the rest like in line 55 (it looks compressed)
RE: corrected
Fig 3 and 126, please make the L- two fontsize smaller than the rest of the text
RE: corrected
line 159, i think it is arginase
RE: corrected
line 163 In some cases like here the whole name was given as in text citation (Hai Du). Another example is line 360
Re: Corrected
Line 174. I think this is the first mention of lactic acid bacteria but further on on LAB is used. PLease abbreviate it here so readers will know what is meant.
RE: corrected
take care that all the gene names are in cursive like in line 260
RE: done
Some references are not according to the guidelines like 9, 22, 77,93
RE: done
Reviewer 2 Report
This manuscript provides useful information on ethyl carbamate, specifically its formation, human risk, and techniques to reduce this compound. Before being considered for further steps, some points need to be clarified:
Abstract: Page 1, Line 18: please correct "e".
Introduction: Page 2, Lines 54 - 68: As the authors mentioned, ethyl carbamate's content varies in different food products. The factors that affect the ethyl carbamate content in food products should be proposed at this point.
Page 4, Figure 2: the names of key enzymes involved in ethyl carbamate formation should be added.
Page 4, Lines 140 - 143: In which conditions (temperatures, concentrations, etc.) can ethyl carbamate form in distilled ethyl alcohol.
Page 6, Lines 212 and 217: the full name of "Lactobacillus" should be mentioned for the first time, and the abbreviation "L. " after that. Please correct this throughout the manuscript.
Page 8, Lines 329-333: It is not clear that the content of ethyl carbamate during exposure and storage can increase significantly. In which exposure or storage conditions can ethyl carbamate increase? Furthermore, in which distillation conditions can ethyl carbamate increase?
Is there any correlation between some metabolite compounds, such as glycerol, and the formation of ethyl carbamate in wine or other alcoholic beverages? If so, please provide such information in the manuscript.
Conclusion: It is better to divide the main winemaking process and propose the technique to prevent the formation of ethyl carbamate in each main process accordingly.
Author Response
Abstract: Page 1, Line 18: please correct "e".
RE: Corrected
Introduction: Page 2, Lines 54 - 68: As the authors mentioned, ethyl carbamate's content varies in different food products. The factors that affect the ethyl carbamate content in food products should be proposed at this point.
RE: the sentence about bread baking was modified. The content of ethyl carbamate in different food products depends on the content of its precursors. The main source of ethyl carbonate in fermented foods and drinks is the reaction between urea and ethanol, and the high ethyl carbonate content in stone fruit spirits is mainly due to the cyanogenic glycosides present in their pits. Since the mechanism of ethyl carbamate formation is described in section 3, we have provided the corresponding description there (lines 117-119).
Page 4, Figure 2: the names of key enzymes involved in ethyl carbamate formation should be added.
RE: added
Page 4, Lines 140 - 143: In which conditions (temperatures, concentrations, etc.) can ethyl carbamate form in distilled ethyl alcohol.
RE: added
Page 6, Lines 212 and 217: the full name of "Lactobacillus" should be mentioned for the first time, and the abbreviation "L. " after that. Please correct this throughout the manuscript.
RE: corrected
Page 8, Lines 329-333: It is not clear that the content of ethyl carbamate during exposure and storage can increase significantly. In which exposure or storage conditions can ethyl carbamate increase? Furthermore, in which distillation conditions can ethyl carbamate increase?
RE: This section was rewritten
Is there any correlation between some metabolite compounds, such as glycerol, and the formation of ethyl carbamate in wine or other alcoholic beverages? If so, please provide such information in the manuscript.
RE: No such correlation was found for glycerol. The effect of glucose and fructose as inhibitors of EC formation in model media was described
Conclusion: It is better to divide the main winemaking process and propose the technique to prevent the formation of ethyl carbamate in each main process accordingly.
RE: corrected
Reviewer 3 Report
The work "Ethyl carbamate in fermented food products: sources of appearance, hazards and methods for reducing its content" provides an important review of the topic, addressing the main aspects of the presence of ethyl carbamate in food and beverages.
Below are my contributions:
- Figure 3 was not cited in the text.
- Lines 208-209: "An alternative way to prevent the formation of ethyl carbamate in wine is the use of lactic acid bacteria Lactobaccilus hilgardii and Oenococcus oeni...", I suggest that the authors mention how these microorganisms were used in the works mentioned, were they introduced as starter cultures?
- Lines 241-242: "Despite numerous and successful attempts to improve the characteristics of wine yeast strains by genetic engineering, only two of them are officially registered for us...", describe which strains these are, giving a brief characterization, place of isolation.
- It would be important to talk about the limits of ethyl carbamate recommended by food and drink legislation, addressing the main differences between products and countries in a succinct way.
- I suggest an item on the main techniques used to detect ethyl carbamate.
- In the conclusion, discuss the prospects of the work, which techniques are most promising for determination, considering for example the cost/benefit ratio.
Author Response
- Figure 3 was not cited in the text.
RE: added
- Lines 208-209: "An alternative way to prevent the formation of ethyl carbamate in wine is the use of lactic acid bacteria Lactobaccilus hilgardii and Oenococcus oeni...", I suggest that the authors mention how these microorganisms were used in the works mentioned, were they introduced as starter cultures?
RE: The presence of these bacteria in the starter cultures reduced the ethyl carbamate content in the fermented food products [61]. In the second work, LAB was used as starter cultures for malolactic fermentation after the end of alcoholic fermentation [15].
- Lines 241-242: "Despite numerous and successful attempts to improve the characteristics of wine yeast strains by genetic engineering, only two of them are officially registered for us...", describe which strains these are, giving a brief characterization, place of isolation.
RE: Information was added. Strain ECMo01 is described in details. The seconds strain ML01 contained the malolactic fermentation cassette.
- It would be important to talk about the limits of ethyl carbamate recommended by food and drink legislation, addressing the main differences between products and countries in a succinct way.
RE: This issue is discussed in the introduction
- I suggest an item on the main techniques used to detect ethyl carbamate.
RE: We added section 6.
- In the conclusion, discuss the prospects of the work, which techniques are most promising for determination, considering for example the cost/benefit ratio.
RE: Conclusion section was corrected